# Suicide Thoughts and Attempts in the Norwegian General Population during the Early Stage of the COVID-19 Outbreak

**DOI:** 10.3390/ijerph18084102

**Published:** 2021-04-13

**Authors:** Tore Bonsaksen, Laila Skogstad, Trond Heir, Øivind Ekeberg, Inger Schou-Bredal, Tine K. Grimholt

**Affiliations:** 1Faculty of Social and Health Sciences, Inland Norway University of Applied Sciences, 2418 Elverum, Norway; 2Faculty of Health Studies, VID Specialized University, 4306 Sandnes, Norway; 3Department of Research, Sunnaas Rehabilitation Hospital HF, 1453 Bjørnemyr, Norway; uxlask@sunnaas.no; 4Department of Nursing and Health Promotion, Faculty of Health Sciences, Oslo Metropolitan University, 0130 Oslo, Norway; 5Norwegian Center for Violence and Traumatic Stress Studies, 0484 Oslo, Norway; trond.heir@medisin.uio.no; 6Institute of Clinical Medicine, University of Oslo, 0450 Oslo, Norway; 7Division of Mental Health and Addiction, Oslo University Hospital, 0424 Oslo, Norway; oeekeber@online.no; 8Faculty of Medicine, University of Oslo, 0372 Oslo, Norway; i.s.bredal@medisin.uio.no; 9Faculty of Health Studies, VID Specialized University, 0370 Oslo, Norway; tine.grimholt@vid.no; 10Department of Acute Medicine, Oslo University Hospital, 0424 Oslo, Norway

**Keywords:** coronavirus, pandemic, population survey, suicidal behavior, suicide

## Abstract

The aim of the study was to examine the prevalence of suicide thoughts and attempts during the early stage of the COVID-19 outbreak and examine pandemic-related factors associated with suicide thoughts in the general Norwegian population. A sample of 4527 adults living in Norway were recruited via social media. Data related to suicide thoughts and attempts, alcohol use, pandemic-related concerns, and sociodemographic variables were collected. Associations with suicide thoughts were analyzed with logistic regression analysis. In the sample, 3.6% reported suicide thoughts during the last month, while 0.2% had attempted suicide during the same period. Previous suicide attempts (OR: 11.93, *p* < 0.001), lower age (OR: 0.69, *p* < 0.001), daily alcohol use (OR: 3.31, *p* < 0.001), being in the risk group for COVID-19 complications (OR: 2.15, *p* < 0.001), and having economic concerns related to the pandemic (OR: 2.28, *p* < 0.001) were associated with having current suicide thoughts. In addition to known risk factors, the study suggests that aspects specific to COVID-19 may be important for suicidal behaviors during the pandemic.

## 1. Introduction

Suicide and suicide attempts convey unbearable personal suffering, and suicides give rise to considerable, long-lasting suffering for family and close ones [1,2,3]. At the population level, the rate of suicide in a country may be considered a crude indicator of the population’s mental health. In Norway, between 600 and 700 persons kill themselves each year [4], and it is suggested that the number of suicide attempts is approximately ten times higher [5]. While the current population suicide rate is somewhat lower compared with the peak in 1988 [6], it has been mostly unchanged since the mid-1990s [7] and appears to have remained similar to other countries in Western Europe and North America (approximately 12–13 deaths per 100,000 inhabitants) [6]. Mental health policies in Norway continue to aim to reduce the population suicide rate [8].

While having thoughts about suicide is far more common than performing actual or attempted suicide, having suicide thoughts has been found to predict subsequent suicide attempts [9,10]. Further, a study in Spain followed up 1241 individuals following a first-time suicide attempt, and found that 20% of them re-attempted suicide at least once, supporting the notion that previous suicidal ideation and behaviors are major risk factors for subsequent suicide attempts [11]. In the USA in 2008–2009, more than eight million people over 18 years of age (3.7% of the adult population) reported having suicidal thoughts during the last year, while approximately one million people (0.5% of the adult population) reported having attempted suicide in the last year [1]. While most suicides are performed by men [6,12,13], suicide thoughts and non-fatal suicidal behaviors are more often found among women. Suicide thoughts are also more commonly found in persons of young age and persons with mental health problems [9]. Mood disorders, and depression in particular, are strongly linked with suicidal behavior, but other mental health problems, such as anxiety disorders, substance use disorders, and impulse control disorders, have also been found to increase the risk of suicide thoughts and attempts [13,14,15,16].

In Norway, the COVID-19 outbreak instigated an immediate lockdown of society [17,18,19], with restrictions that were gradually lifted over the following months. Schools and universities were closed, as were non-essential workplaces and businesses such as hairdressers and physiotherapy clinics. Working from home became the new standard for many groups of employees for whom this was a possibility, while others lost their jobs or were temporarily furloughed [20]. Flights and travel were canceled, as were all cultural events requiring physical attendance, and the public were generally encouraged to stay at home and maintain a secure physical distance to people from outside the household.

The relatively small number of infected (*n* = 8412) and diseased (*n* = 244, 2.9% of the infected) people in Norway during the first three months of the pandemic [21] may be interpreted as evidence of successful handling of the crisis during the early stage. The majority of deaths was among people over 70 years of age with at least one chronic disease [21]. However, in view of the severe restrictions in people’s social life and a sharp increase in unemployment, much research in Norway and internationally has been concerned with the population’s mental health in the pandemic context. A meta-analysis of studies published before May 2020 found high prevalence rates of stress (29.6%), anxiety (31.9%), and depression (33.7%) in general population studies [22]. With regard to suicidal ideation, a three-wave study from the UK found that significantly more participants had suicide thoughts during the second (9.2%) and third waves (9.8%) of data collection, compared to the first wave (8.2%), thus indicating a worrying time trend [23]. In a recent Norwegian study, however, the increase in point estimates of suicide thoughts in the population from before (3.2%) to after the COVID-19 outbreak (4.1%–4.2%) was not statistically significant [24].

Changes in the rates of and risk factors for suicidal behavior in the population have important implications for the targeting and planning of mental health care services. Increased knowledge in this area may assist more targeted initiatives for suicide prevention. Previous studies have found that potentially stressful aspects specific to COVID-19, such as having economic concerns, experiencing isolation or quarantine, and perceiving to have high risk of complications, were associated with higher risk of mental health problems [25,26,27]. Although reviews have found weak evidence of increased suicidal behavior during previous emerging viral outbreaks [28], recent studies suggest that mental health problems combined with a long-lasting stressful life experience, such as the COVID-19, may constitute a particularly problematic situation that can have an impact on suicide thoughts and behaviors in the population [29,30].

### Study Aim

The aims of the study were to examine the prevalence of suicide thoughts and attempt during the early stage of the COVID-19 outbreak and examine pandemic-related factors associated with suicide thoughts in the general Norwegian population.

## 2. Materials and Methods

### 2.1. Design

The CORONAPOP study, a population-based cross-sectional survey, was open to all citizens between 8 April 2020 and 20 May 2020. Several institutions, including Oslo University Hospital, Sunnaas Hospital, University of Oslo, and Oslo Metropolitan University, hosted and disseminated the web-link to the survey. The link to the survey was openly disseminated on social media platforms, such as Facebook, Twitter, LinkedIn, and Instagram, by the individual researchers and other individuals. The study was also featured in national and local newspapers.


### 2.2. Sample

Norwegian citizens aged 18 years or older were invited to participate. There were no exclusion criteria.


### 2.3. Measures

Through the web-based survey, sociodemographic and health-related data were collected as self-report measures. Several of the employed measures were identical to the ones used in the Norwegian population health survey (NORPOP), which was conducted as a postal survey from 2014–2015 [31,32,33,34]. 

#### 2.3.1. Sociodemographic Variables

Data were collected for age group (18–29 years, 30–39 years, 40–49 years, 50–59 years, 60–69 years, and 70 years or older) gender (male/female), highest completed education level (elementary school, high school, less than four years of higher education, and four years of higher education or more), employment status (working/in education versus not), cohabitation status (living with spouse or partner versus not), and size of place of residence (<200 inhabitants, 200–19,999 inhabitants, 20,000–99,999 inhabitants, 100,000 inhabitants or more). A significant part of the Norwegian population lives in rural areas. The categories were constructed based on the consideration that about 20–30% of the population belongs to each of the categories, and the NORPOP study used the same categories.

#### 2.3.2. Suicide Thoughts and Attempts

We used the phrase: “Below is a list of health problems. Do you have, or have you had, any of these?” Among the listed problems were suicide thoughts and suicide attempts. The response alternatives were “no”, “yes previously, but not during the last month” and “yes, during the last month”. Those who confirmed having suicide thoughts and/or suicide attempt during the last month were classified as currently having the relevant mental health problems. Those who confirmed having suicide thoughts and/or suicide attempt “yes previously, but not during the last month” were classified as having had the relevant mental health problem previously.

#### 2.3.3. Alcohol Use

We used the phrase: “Use of alcohol and addictive drugs and pharmaceuticals: have you used any of these?” Below was a list containing alcohol and other substances. Response options were “no”, “sometimes”, “weekly”, and “daily”. Participants who reported that they used alcohol daily were classified as daily drinkers.

#### 2.3.4. Problems Related to the Pandemic

Relating to the COVID-19 situation, participants were asked to respond “yes” or “no” to the following questions in relation to the pandemic: (a) “Do you have economic concerns?”, (b) “Have you been in quarantine or in isolation due to the coronavirus?”, and (c) “Are you in the risk group for complications if infected by COVID-19?”

### 2.4. Statistical Analysis

Frequencies and proportions (%) were calculated for all categorical variables, and all sociodemographic variables were cross tabulated with the reported suicide thoughts and attempts. Single predictor and multiple predictor logistic regression analyses were performed to assess associations between previous suicide attempts, sociodemographic variables (age and gender), alcohol use, COVID-19-related problems (risk group, economic concerns, and having experienced quarantine/self-isolation), and current suicide thoughts. The odds ratio (OR) with 95% confidence interval (CI) was reported. IBM SPSS Statistics version 26 [35] was used for statistical analyses, and due to the large sample size, the significance level was set at 1%.

### 2.5. Ethics

The questionnaires were answered anonymously. Ethical approval for conducting the study was granted from the Regional Committee for Medical and Healthcare Ethics (REK no. 130447).


## 3. Results

### 3.1. Sample Characteristics

The sociodemographic characteristics of the sample (*n* = 4527) are displayed in Table 1. Over half of the sample was below 40 years of age. The majority were women (85.0%), had higher education (75.5%), and were employed or enrolled in education (81.0%). With respect to urbanity, the largest group of participants lived in cities with more than 100,000 inhabitants (46.3%).

While there was substantial overlap between those with current suicide thoughts and previous suicide attempts (*n* = 45, 28.0% of those with current suicide thoughts), there was only marginal overlap between those with current and previous suicide thoughts (*n* = 5, 3.1% of those with current suicide thoughts). Due to very unequal group sizes, the logistic regression analyses therefore included previous suicide attempt as predictor, but not previous suicide thoughts. In the sample, 161 (3.6%) reported to have had suicide thoughts during the last month, while seven (0.2%) had attempted suicide. Among those with suicide thoughts during the last month, 4.3% reported that they also had attempted suicide in the same period. In addition, 138 (3.0%) reported daily use of alcohol, 1061 (23.4%) reported to be in the risk group, 985 (21.8%) reported to have economic concerns, and 1278 (28.2%) reported to have been in quarantine or self-isolation due to COVID-19.

### 3.2. Factors Associated with Current Suicide Thoughts

The results from the single and multiple logistic regression analysis are reported in Table 2. The unadjusted analyses showed that a previous suicide attempt was associated with a 15 times higher likelihood of having current suicide thoughts, compared to no previous suicide attempt. In addition, lower age, daily alcohol use, considering oneself to be in the risk group, and having economic concerns were associated with current suicide thoughts. In the adjusted model, all of these variables remained significantly associated with the outcome: previous suicide attempt (OR: 11.93, *p* < 0.001), lower age (OR: 0.69, *p* < 0.001), daily alcohol use (OR: 3.31, *p* < 0.001), considering oneself to be in the risk group (OR: 2.15, *p* < 0.001), and having economic concerns (OR: 2.28, *p* < 0.001).

## 4. Discussion

### 4.1. Summary of Results

This study examined the prevalence of suicide thoughts and attempts during the early stage of the COVID-19 outbreak and examined factors associated with current suicide thoughts in the general Norwegian population. In this sample, 161 participants (3.6%) reported having had suicide thoughts during the last month, while seven (0.2%) had attempted suicide. In the fully adjusted model, previous suicide attempt, lower age, daily alcohol use, risk of complications and having economic concerns were associated with higher odds of current suicide thoughts.

### 4.2. Suicide Thoughts and Attempts

The prevalence of suicide thoughts during the last month (3.6%) reported in this early stage COVID-19 study is similar to the prevalence reported in a previous general population study from Norway (4.1–4.2%) [24], while it is lower than the prevalence found around the same time in the UK (8.2–9.8%) [23] and the USA (10.7%) [30]. Considering that the prevalence rate of suicide thoughts in this study was similar to the rate reported in another Norwegian study conducted during the same time [24], the validity of our findings is supported. The differences in comparison to other countries may reflect general cross-national differences in suicidal ideation, considering that large differences in lifetime prevalence have been found between countries and regions [9,13]. Possibly, the differences may also reflect differences related to the impact of COVID-19 between countries. No doubt, the UK and USA have had much larger numbers of infected and dead than Norway [36]. Infection rates and the number of citizens hospitalized or dead, but also burdens related to the general pandemic situation (restrictions on meeting people, worry about family and friends, financial concerns, etc.), may translate into higher rates of suicide thoughts in those countries most strongly affected by COVID-19.

Among those with suicide thoughts during the last month, 4.3% reported that they also had attempted suicide in the same period. Suicide thoughts have been found to be strongly related to suicide attempts during a person’s lifetime [37]. Based on cross-national data from 17 countries, Nock and colleagues [9] reported that more than 60% of all suicide attempts occur within one year after the onset of suicide thoughts. Considering these findings, our study’s ratio of those having made a suicide attempt among those who currently have suicide thoughts seems low. We considered that having suicide thoughts might be a longstanding problem among a proportion of the sample, knowing that many people live with such thoughts over time without progressing onto suicide attempts. However, this interpretation was not supported, given the only marginal overlap between previous and current suicide thoughts. This may imply that the vast majority of those reporting current suicide thoughts did not have this problem previously. Alternatively, or in combination with the former, some of those indicating current suicide thoughts may have felt that their current situation was most pressing, and while reporting current suicide thoughts they may have failed to indicate previous suicide thoughts. In any case, for persons with suicide thoughts first arising during the early stage of COVID-19, one may be particularly concerned about their development and whether they will translate into suicide attempts at a later stage.

### 4.3. Factors Associated with Suicide Thoughts

Having attempted suicide previously was strongly associated with reporting current suicide thoughts. This finding corresponds with previous studies, in which results have indicated that suicidal behaviors in the past predict suicide attempts [11] and suicide [38] at a later point in time. Moreover, lower age and using alcohol daily were associated with higher odds of current suicide thoughts. These results are in concert with findings from other general population studies. For example, Nock and colleagues [9] reported that younger age and having a mental disorder were consistent predictors of suicide thoughts, plans, and attempts. Moreover, during the COVID-19 pandemic, suicidal ideation has also been found to decrease progressively with age [30]. Mental disorders such as anxiety [39,40] and depression [41,42] have often been linked with higher risk of suicide thoughts, and use of alcohol—i.e., higher drinking frequency, higher drinking quantity, and binge drinking—has also been found to increase the risk [43,44].

With a view to the COVID-19 situation, the study showed that risk of complications from contracting the virus and having economic concerns related to the pandemic were associated with higher risk of having suicide thoughts. Both factors have been found previously to be related to high stress levels corresponding to symptom-defined PTSD [25], and a pathway from risk group and economic concerns via high stress and anxiety to suicide thoughts, is viable. A pathway to suicide thoughts may also go via depression, as self-identifying as being in the risk group and having economic concerns have been found to be associated with higher risk of depression [26].

### 4.4. Study Limitations

The cross-sectional survey makes it impossible to establish causal relationships between variables. The study is therefore limited in its mere detection of factors statistically associated with suicide thoughts. While a range of variables were used as possible predictors of suicide thoughts, the inclusion of additional variables might well have improved the model. For example, anger, impulsivity, worries about one’s physical health, and suicidality have been shown to be higher among individuals with psychiatric illnesses compared to healthy controls [45]. Thus, factors such as pandemic-related anger and worries about physical health might also be associated with suicide thoughts and suicide attempts during the COVID-19 crisis.

The recruitment strategy was based on disseminating the link to the survey via various social media. As a result, the self-selected study sample was dominated by young, urban, and highly educated persons, and the vast majority were female. Possibly, respondents might have felt more affected by the pandemic situation, compared to those opting not to participate in the study. The same pattern of sample skewness has been shown in COVID-19 studies from several countries [17,46,47]. Generalization of the findings to the national population should therefore be done with caution. Prevalence rates must be interpreted with caution, while associations with risk factors are more likely to be universal [9].

Standardized and well-tested mental health measures are often preferred over single-item scales for assessing mental health problems, and future studies may indeed include the use of standardized measures. However, standardized measures are generally longer and place a larger burden on participants, especially when used as part of a more comprehensive survey. Moreover, studies have demonstrated that the use of single-item measures can be a valid and reliable option for measuring mental health variables [48,49,50,51]. Additionally, assessing whether a person has had suicide thoughts and attempts during a given period is relatively straightforward, and for this purpose we believe the chosen questionnaire items were appropriate. However, we cannot rule out the possibility that those indicating current suicide thoughts may have failed to indicate previous suicide thoughts.

## 5. Conclusions

Among those responding to the survey, 3.6% reported suicide thoughts during the last month and 0.2% reported suicide attempts during the same period. In addition to previous suicide attempts, younger age, daily use of alcohol, being in the risk group of complications if infected by COVID-19, and experiencing economic concerns during the pandemic outbreak were associated with having suicide thoughts. The study supports previous findings concerned with age and alcohol use in relation to suicide thoughts and suggests that specific aspects of the pandemic situation may be important for suicidal behaviors during the pandemic.

## Figures and Tables

**Table 1 ijerph-18-04102-t001:** Sociodemographic characteristics of the sample, and suicide thoughts and suicide attempts in sample subgroups.

Characteristics	Total Sample	Suicide Thoughts		Suicide Attempts	
		Last Month	Previously	Last Month	Previously
	*n* (%)	*n* (%)	*n* (%)	*n* (%)	*n* (%)
All	4527 (100.0)	161 (3.6)	479 (10.6)	7 (0.2)	153 (3.4)
Age group					
18–29	1156 (25.5)	71 (6.1)	193 (16.7)	4 (0.3)	63 (5.4)
30–39	1220 (26.9)	48 (3.9)	146 (12.0)	0 (0.0)	43 (3.5)
40–49	931 (20.6)	19 (2.0)	79 (8.5)	2 (0.2)	18 (1.9)
50–59	766 (16.9)	16 (2.1)	47 (6.1)	1 (0.1)	16 (2.1)
60–69	354 (7.8)	7 (2.0)	10 (2.8)	0 (0.0)	11 (3.1)
70 or above	100 (2.2)	0 (0.0)	4 (4.0)	0 (0.0)	2 (2.0)
Gender ^a^					
Male	659 (14.6)	17 (2.6)	57 (8.6)	2 (0.3)	14 (2.1)
Female	3850 (85.0)	143 (3.7)	422 (11.0)	5 (0.1)	139 (3.6)
Highest completed education ^b^					
Elementary school	591 (13.1)	30 (5.1)	92 (15.6)	1 (0.2)	43 (7.3)
High school	514 (11.4)	32 (6.2)	72 (14.0)	3 (0.6)	32 (6.2)
Higher education < 4 years	1376 (30.4)	55 (4.0)	147 (10.7)	1 (0.1)	38 (2.8)
Higher education ≥ 4 years	2041 (45.1)	44 (2.2)	168 (8.2)	2 (0.1)	40 (2.0)
Employment					
Employed or in education	3667 (81.0)	99 (2.7)	348 (9.5)	4 (0.1)	92 (2.5)
Not employed and not in education	860 (19.0)	62 (7.2)	131 (15.2)	3 (0.3)	61 (7.1)
Cohabitation status					
Living with spouse or partner	2714 (60.0)	44 (1.6)	265 (14.6)	2 (0.1)	53 (2.0)
Not living with spouse or partner	1813 (40.0)	117 (6.5)	214 (7.9)	5 (0.3)	100 (5.5)
Size of place of residence ^c^					
Rural	187 (4.1)	4 (2.1)	20 (10.7)	0 (0.0)	6 (3.2)
Village	1141 (25.2)	34 (3.0)	123 (10.8)	3 (0.3)	37 (3.2)
Town	1091 (24.1)	30 (2.7)	113 (10.4)	2 05.2)	35 (3.2)
City	2098 (46.3)	93 (4.4)	222 (10.6)	2 (0.1)	75 (3.6)

Note. ^a^ Eighteen participants (0.4%) did not state gender, ^b^ 5 participants (0.1%) did not state education level, ^c^ 10 participants (0.2%) did not state size of place of residence. Rural is <200 inhabitants, village is 200–19,999 inhabitants, town is 20,000–99,999 inhabitants, and city is >100,000 inhabitants.

**Table 2 ijerph-18-04102-t002:** Associations with suicide thoughts last month (*n* = 4493).

Independent Variables	Unadjusted			Adjusted		
	OR	99%CI	*p*	OR	99%CI	*p*
Previous suicide attempt	15.30	9.12–25.66	<0.001	11.93	6.93–20.54	<0.001
Higher age group	0.67	0.55–0.80	<0.001	0.69	0.56–0.84	<0.001
Female gender	1.46	0.75–2.85	0.15	1.26	0.61–2.60	0.40
Daily alcohol use	2.98	1.36–6.50	<0.001	3.31	1.40–7.86	<0.001
Risk group	1.68	1.08–2.61	<0.01	2.15	1.31–3.52	<0.001
Economic concerns	3.22	2.12–4.89	<0.001	2.28	1.45–3.60	<0.001
Quarantine/isolation	0.95	0.60–1.52	0.80	0.80	0.49–1.31	0.25
Model Fit					*p* < 0.001	
Nagelkerke R^2^					17.4%	

Note. Dependent variable is having had suicide thoughts during the last month. Age group is ten-year intervals. Risk group is based on self-classification. Economic concerns are concerns related to the pandemic situation. Quarantine/isolation refers to having been in quarantine or self-isolation due to COVID-19.

## Data Availability

The data presented in this study are available on request from the corresponding author by completion of the research project. The data are not publicly available due to ongoing publication from the project.

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
