# Peer review of "Suicide Thoughts and Attempts in the Norwegian General Population during the Early Stage of the COVID-19 Outbreak"

_ijerph, 2021, doi:10.3390/ijerph18084102_

Round 1
Reviewer 1 Report
The aim of the study was to examine the prevalence of suicide thoughts and -attempts 17 during the early stage of the COVID-19 outbreak and examine factors associated with suicide 18 thoughts in the general Norwegian population. This is a timely and important topic. Unfortunately, the methodology and discussion of the data does not meet publishable standards. Background: It would be helpful if the authors would give some pertinent background about Covid affected Norway. What were the lockdown measures? Economic supports? Infection rates? Study design: This is a fishing expedition. There was no stated hypothesis, nor any attempt to account for the large sample size (e.g., Bonferroni, effect size calculation) It would be helpful to see the actual wording of the suicidal ideation question since there is a lot of research showing the effect of wording in response in this area. Discussion: The results show that suicidal ideation and suicide attempt rates were no different than pre-Covid numbers. I do not follow the "The similar rates 182 reported across the Norwegian studies conducted during the same time strengthen our findings" statement.Author Response
Authors: Thank you for the comments and suggestions to the manuscript, which we believe have served to improve its quality. All issues raised by the reviewer have been addressed point by point below, and all changes in the manuscript are performed using track changes for Word. We look forward to hearing from you.
***************************************************************************
Reviewer 1
R1: The aim of the study was to examine the prevalence of suicide thoughts and -attempts during the early stage of the COVID-19 outbreak and examine factors associated with suicide thoughts in the general Norwegian population. This is a timely and important topic. Unfortunately, the methodology and discussion of the data does not meet publishable standards.
Authors: No response required.
R1: Background: It would be helpful if the authors would give some pertinent background about Covid affected Norway. What were the lockdown measures? Economic supports? Infection rates?
Authors: More information about how the pandemic affected, and was handled in, Norway is included in the revised introduction section. Economic supports from the government were introduced after the data collection for this study was completed and are therefore not relevant for the interpretation of the results.
R1: Study design: This is a fishing expedition. There was no stated hypothesis, nor any attempt to account for the large sample size (e.g., Bonferroni, effect size calculation).
Authors: The research project as a whole aimed to assess reactions to the COVID-19 in the Norwegian population. Outcomes included suicide thoughts and attempt, in addition to a range of other variables. Therefore, we disagree that the study reflects a “fishing expedition”; it is rather one of several studies about reactions in the population during the early stage of the pandemic outbreak. It is true that no specific hypothesis was stated; the study is explorative. In view of the radically new social circumstances imposed by COVID-19, we believe explorative studies are warranted. In the revised analyses, we have used p < 0.01 as the threshold for statistical significance, to account for the large sample size (see section 2.4 and Table 2).
R1: It would be helpful to see the actual wording of the suicidal ideation question since there is a lot of research showing the effect of wording in response in this area.
Authors: The wording of the suicidal ideation question has been clarified in the relevant Measures section (2.3.2).
R1: Discussion: The results show that suicidal ideation and suicide attempt rates were no different than pre-Covid numbers. I do not follow the "The similar rates reported across the Norwegian studies conducted during the same time strengthen our findings" statement.
Authors: We have clarified the relevant sentence (see Discussion section).
Reviewer 2 Report
The study is very important and necessary. In my opinion it may be of interest to the readers of this journal. Your design is correct. However, there is my main criticism. In my opinion, the authors should present why they did not use a previously validated survey?
My second criticism: Authors must show the consistency validation values of their questionnaire questions used.
Author Response
Authors: Thank you for the comments and suggestions to the manuscript, which we believe have served to improve its quality. All issues raised by the reviewer have been addressed point by point below, and all changes in the manuscript are performed using track changes for Word. We look forward to hearing from you.
***************************************************************************
Reviewer 2
R2: The study is very important and necessary. In my opinion it may be of interest to the readers of this journal. Your design is correct. However, there is my main criticism. In my opinion, the authors should present why they did not use a previously validated survey?
Authors: While the use of standardized measures are often preferred over single-item scales, one should also consider the complexity of the studied concept and the context within which the concept is assessed. We argue that the measurement of suicide thoughts and -attempt within a given time frame is relatively straightforward, and that the given context (a relatively large survey) warrants the use of single-item measures of these concepts. See also additions about measurement in the limitations section.
R2: My second criticism: Authors must show the consistency validation values of their questionnaire questions used.
Authors: Internal consistency is an important aspect of the reliability of measures used in research. However, measuring the internal consistency between items requires the use of measures with several items. No such measures were used in this study (see Measures section, 2.3).
Reviewer 3 Report
Reviewer’s Comments
The manuscript “Suicide Thoughts and Attempts in the Norwegian General Population during the Early Stage of the COVID-19 Outbreak” describes a study of suicidal thought in N=4527 adults, recruited through social media. In this convenience sample, 3.6% reported suicidal thoughts. Logistic regression was used to test risk factors or groups of risk factors. This is a simple study, with some interesting results, but its significance is diminished because of the nature of the self-selected sample. The Discussion and Limitations sections do not emphasize the limitations imposed by the nature of the self-selected sample, a discussion of who is likely to respond to such surveys and how that may bias the results would be beneficial. The choice of the risk factors tested with logistic regression models seems rather arbitrary, given the data presented in the Demographic table shows other differences in the rate of suicidal thought. If more tests were performed and only some are presented, that should be reported and adjusted for. Not adjusting for previous suicide attempt and previous suicide thought specifically, only for self-reported “Risk Group”, is a missed opportunity, as it could help more clearly identify factors that have predictive power for Covid-related suicidality. Perhaps the overlap between past and current thought/attempt was such that it precludes an adjustment- if so, that should be stated. There is a curious lack of reporting and discussion on the rellationship of past vs. current suicidality, although it must have been considered at the design of the study, since the questions were asked and answers reported.
Specific Comments:
- Statistical terminology: “Single and multiple logistic regressions were used…” should be “Single predictor and multiple predictor logistic regressions…”
- Other terminology: the use of the term “(attempted to) commit suicide” is discouraged by the International Association for Suicide Research and other organizations, as it carries connotations of criminality. Instead, one can say “attempted to kill themselves”, “attempted to end their lives”, or simply “had a suicide attempt”.
- How many single-predictor tests were performed? Risk factors’ significance levels should be adjusted for multiple testing in the single predictor models, unless they are used solely for the purpose of screening for the multi-predictor model. Table 1 includes many factors that are not mentioned in the Results, were they all tested? If not, why not? Most of these (employment and living situation, rural/urban region) could have been risk factors.
Author Response
Authors: Thank you for the comments and suggestions to the manuscript, which we believe have served to improve its quality. All issues raised by the reviewer have been addressed point by point below, and all changes in the manuscript are performed using track changes for Word. We look forward to hearing from you.
***************************************************************************
Reviewer 3
R3: The manuscript “Suicide Thoughts and Attempts in the Norwegian General Population during the Early Stage of the COVID-19 Outbreak” describes a study of suicidal thought in N=4527 adults, recruited through social media. In this convenience sample, 3.6% reported suicidal thoughts. Logistic regression was used to test risk factors or groups of risk factors.
Authors: No response required.
R3: This is a simple study, with some interesting results, but its significance is diminished because of the nature of the self-selected sample.
Authors: We agree that the self-selected sample, non-representative of the Norwegian general population, is a significant limitation. However, we would argue that the large sample size and our ability to assess suicide thoughts in relationship to variables directly linked to the pandemic situation, makes the study interesting. See revised limitations section.
R3: The Discussion and Limitations sections do not emphasize the limitations imposed by the nature of the self-selected sample, a discussion of who is likely to respond to such surveys and how that may bias the results would be beneficial.
Authors: We believe that the place for methodological concerns are in the limitations section. We have added to the discussion about the self-selected sample and who is likely to participate in such surveys; see revised limitations section.
R3: The choice of the risk factors tested with logistic regression models seems rather arbitrary, given the data presented in the Demographic table shows other differences in the rate of suicidal thought. If more tests were performed and only some are presented, that should be reported and adjusted for.
Authors: Table 1 was meant to be descriptive; therefore, no tests of group differences were performed. The aim of the study was to examine suicide thoughts and -attempt and assess suicide thoughts in relationship to pandemic-related variables (clarified in the study aim section; section 1.1). Thus, the variables included in the logistic regression model were based on the reasoning put forward throughout the introduction section. Throughout the introduction, we argue that age, gender, and substance use are relevant correlates to suicide thoughts. Moreover, we emphasize the relevance of exploring pandemic-related variables as risk factors for suicide thoughts, as the current study was conducted during the early stage of the COVID-19 pandemic.
R3: Not adjusting for previous suicide attempt and previous suicide thought specifically, only for self-reported “Risk Group”, is a missed opportunity, as it could help more clearly identify factors that have predictive power for Covid-related suicidality. Perhaps the overlap between past and current thought/attempt was such that it precludes an adjustment- if so, that should be stated.
Authors: Additional analyses showed that while previous suicide attempt was strongly associated with current suicide thoughts, previous suicide thoughts was strongly and inversely related to current suicide thoughts. That is, while there was substantial overlap between those with previous suicide attempt and current suicide thoughts (n=45, 28 % of those with current suicide thoughts), there was practically no overlap between those with previous and current suicide thoughts (n=5, 3.1 % of those with current suicide thoughts). In the revised multivariate analysis, we therefore adjusted for previous suicide attempt but not previous suicide thoughts (due to very unequal group sizes). The inclusion of previous suicide attempt in the multiple logistic regression did not significantly change the other associations. See revised Results section (3.1), Discussion (4.2) and Table 2. Please note that the variable ‘risk group’ refers to risk of complications if infected with the coronavirus, and not risk of suicide.
R3: There is a curious lack of reporting and discussion on the relationship of past vs. current suicidality, although it must have been considered at the design of the study, since the questions were asked and answers reported.
Authors: The relationship between past and current suicidal behavior is briefly included in the revised introduction, results (section 3.1) and discussion (section 4.2).
R3: Statistical terminology: “Single and multiple logistic regressions were used…” should be “Single predictor and multiple predictor logistic regressions…”
Authors: Changed as suggested (see section 2.4).
R3: Other terminology: the use of the term “(attempted to) commit suicide” is discouraged by the International Association for Suicide Research and other organizations, as it carries connotations of criminality. Instead, one can say “attempted to kill themselves”, “attempted to end their lives”, or simply “had a suicide attempt”.
Authors: Changed throughout the manuscript, as suggested.
R3: How many single-predictor tests were performed? Risk factors’ significance levels should be adjusted for multiple testing in the single predictor models, unless they are used solely for the purpose of screening for the multi-predictor model.
Authors: For the revised manuscript, seven single-predictor tests were performed (see Table 2). In view of the large sample size and previous reviewer comments, all significance levels have been set at p<0.01.
R3: Table 1 includes many factors that are not mentioned in the Results, were they all tested? If not, why not? Most of these (employment and living situation, rural/urban region) could have been risk factors.
Authors: Table 1 intends to describe the sample and let the reader have some idea of how suicide thoughts and suicide attempt were distributed across sample subgroups. Tests of associations are displayed in Table 2, and the seven predictor variables tested there were the only ones tested. The inclusion of variables is aligned with the study aim and the reasoning put forward in the introduction section – the focus is on pandemic-related variables, in addition to some well-known predictors (age, gender, alcohol use, previous suicide attempt).
Round 2
Reviewer 3 Report
No further comments
Author Response
Thank you for reviewing this manuscript.